# Endocannabinoid System in Hepatic Glucose Metabolism, Fatty Liver Disease, and Cirrhosis

**DOI:** 10.3390/ijms20102516

**Published:** 2019-05-22

**Authors:** Ivonne Bazwinsky-Wutschke, Alexander Zipprich, Faramarz Dehghani

**Affiliations:** 1Department of Anatomy and Cell Biology, Martin Luther University Halle-Wittenberg, Grosse Steinstrasse 52, D-06108 Halle (Saale), Germany; ivonne.bazwinsky@medizin.uni-halle.de; 2Laboratory of Molecular Hepatology, Clinic of Internal Medicine I, Martin Luther University Halle-Wittenberg, Ernst-Grube-Strasse 40, D-06120 Halle (Saale), Germany; alexander.zipprich@medizin.uni-halle.de

**Keywords:** endocannabinoid system, hepatic cannabinoid receptors, G protein-coupled receptor (GPR)119, GPR55, peroxisome proliferator-activated receptor (PPAR)

## Abstract

There is growing evidence that glucose metabolism in the liver is in part under the control of the endocannabinoid system (ECS) which is also supported by its presence in this organ. The ECS consists of its cannabinoid receptors (CBRs) and enzymes that are responsible for endocannabinoid production and metabolism. ECS is known to be differentially influenced by the hepatic glucose metabolism and insulin resistance, e.g., cannabinoid receptor type 1(CB_1_) antagonist can improve the glucose tolerance and insulin resistance. Interestingly, our own study shows that expression patterns of CBRs are influenced by the light/dark cycle, which is of significant physiological and clinical interest. The ECS system is highly upregulated during chronic liver disease and a growing number of studies suggest a mechanistic and therapeutic impact of ECS on the development of liver fibrosis, especially putting its receptors into focus. An opposing effect of the CBRs was exerted via the CB_1_ or CB_2_ receptor stimulation. An activation of CB_1_ promoted fibrogenesis, while CB_2_ activation improved antifibrogenic responses. However, underlying mechanisms are not yet clear. In the context of liver diseases, the ECS is considered as a possible mediator, which seems to be involved in the synthesis of fibrotic tissue, increase of intrahepatic vascular resistance and subsequently development of portal hypertension. Portal hypertension is the main event that leads to complications of the disease. The main complication is the development of variceal bleeding and ascites, which have prognostic relevance for the patients. The present review summarizes the current understanding and impact of the ECS on glucose metabolism in the liver, in association with the development of liver cirrhosis and hemodynamics in cirrhosis and its complication, to give perspectives for development of new therapeutic strategies.

## 1. Introduction

The endocannabinoid system (ECS) is a complex physiological system that affects metabolic pathways in the brain and in peripheral organs [1]. The ECS comprises cannabinoid receptors (CBRs), their endogenous ligands, termed endocannabinoids, and the enzymes involved in the endocannabinoid synthesis and degradation [2,3,4,5].

Cannabinoids are a class of plant derived or synthetic compounds, including the herbal cannabinoids that occur in the cannabis plant (Cannabis sativa) or synthetic analogs that might or might not act on classical CBRs [4]. The endogenously produced counterparts are termed as endocannabinoids. Endocannabinoids are synthesized “on demand” from long-chain polyunsaturated fatty-acids and act on cells in paracrine or autocrine manner [2,6]. The first-discovered and best-studied endocannabinoids are anandamide (N-arachidonoyl-ethanolamine, AEA) and 2-arachidonoylglycerol (2-AG). Newly proposed endocannabinoids such as 2-arachidonyl-glyceryl ether (noladin, 2-AGE), O-arachidonoyl-ethanolamine (virodhamine), or N-arachidonoyl-dopamine have been isolated, although their physiological functions are only partly understood [4]. Anandamide is a ligand for the CBRs type 1 and 2 (CB_1_, CB_2_), however, it is functionally selective for the CB_1_ and can interact with non-classic CBRs such as the transient receptor potential vanilloid type 1 (TRPV1) receptor [7] or the nuclear receptors peroxisome proliferation-activated receptor (PPAR)α and PPARγ at high micromolar concentrations [8,9]. 2-AG is a ligand for the CB_1_ and CB_2_ receptors und is generally produced from the hydrolysis of 2-arachidonate-containing diacylglycerols. AEA synthesis is under the control of a selective calcium-dependent phospholipase D namely N-arachidonoyl-phosphatidylethanolamine-phospholipase D (NAPE-PLD) whereas the fatty-acid amide hydrolase (FAAH) catalyzes the hydrolysis of anandamide, and to a certain extent 2-AG. 2-AG is formed by two diacylglycerol lipases α and β (DAGLα and β) and hydrolyzed by the monoacylglycerol lipase (MAGL). Another way to degrade AEA and 2-AG involve oxidation via cyclooxygenase-2 (COX-2) [10,11,12]. Considering the CBRs, amongst the numerous CBRs, the most investigated are the CB_1_ and CB_2_ receptors which belong to members of the superfamily of G protein-coupled receptors (GPCRs). The CB_1_ receptor (or CNR1) was originally cloned from rat cerebral cortex [13] and its identity as a CBR was subsequently revealed [14]. The CB_2_ receptor (or CNR2) was first cloned from rat spleen [15]. Its distinct actions from the CB_1_ were confirmed after the creation of CB_2_ receptor-deficient mice [16]. Besides the classical CB_1_ and CB_2_ receptors, other deorphanized GPCRs such as GPR3, GPR6, GPR18, GPR55, and GPR119 and non-cannabinoid receptors, like the TRPV1, or the peroxisome proliferator-activated receptors (PPARs), are suggested as part of the endocannabinoid system [5,17].

## 2. Expression of Cannabinoid Receptors CB_1_ and CB_2_

Cannabinoid receptor type 1 (CB_1_) and type 2 (CB_2_) are known to be highly abundant in the central nervous system (CB_1_: [18,19,20], CB_2_: [21,22,23]). Both receptors were also identified in peripheral organs, including the liver [24,25,26,27,28]. Earlier studies have investigated the presence of the ECS in peripheral organs and found conflicting results. Most of these studies denied the presence of CB_1_ and CB_2_ in the non-pathological liver [29,30,31,32,33]. However, more recent studies have demonstrated CB_1_- and CB_2_-mRNA expression in normal human liver tissue [34]. Recently, it is widely accepted that both receptors are present in liver tissue. Accordingly, CB_1_ and CB_2_ mediate a number of biological functions in different types of liver cells, including hepatocytes, stellate cells, and sinusoidal endothelial cells [35]. The main cannabinoid receptor found in liver cells is the CB_1_ receptor. It was detected in hepatic sinusoidal cells, stellate cells and hepatocytes, whereas the CB_2_ receptor was identified in Kupffer cells and stellate cells [36,37,38,39,40]. Interestingly, a novel human CB_1_ receptor isoform (CB_1b_) has been found to be highly expressed in pancreatic β-cells and hepatocytes, but not in the brain [40]. Both CB_1_ and CB_2_ receptors signal through G_i/0_ proteins, inhibit adenylyl cyclase and regulate ion channels e.g., rectifying potassium channels or voltage-gated calcium channels [41], although there is also evidence that CB_1_ can act via G_s_ proteins. The activity of a variety of intracellular kinases, e.g., mitogen activated protein kinases and extracellular signal-regulated kinases (MAPK/ERK pathway), c-Jun N-terminal kinases (JNKs), and protein kinase B (Akt) are regulated by CBRs.

Similar to other GPCRs, the intracellular C-terminal domain of CB_1_ receptor mediates its effects into different signaling pathways, such as protein kinase A (PKA), mitogen-activated protein kinases (MAPKs), etc. The proximal C-terminal domain represented by amino acids 401–417 is essential for protein-protein interactions. The distal C-terminal tail domain represented by amino acids 418–472 determines the magnitude and kinetics of intracellularly induced signals [42,43,44].

### 2.1. Diurnal Expression of Hepatic Cannabinoid Receptors CB_1_ and CB_2_

In view of hepatic CB_1_ and CB_2_ expression, we have been able to show a diurnal expression pattern of CB_1_ and CB_2_ mRNA in liver tissue of young and middle-aged normoglycemic Wistar rats. The diurnal mRNA expression was followed accordingly by comparable translation levels of CB1 receptor protein [45]. The same study revealed that diurnal expression levels of cannabinoid receptors are influenced by the diabetic state of the rats. The animals showed alterations in their diurnal profile of both receptors in streptozotocin-treated 12- and 51-week-old or normoglycemic rats [45]. Interestingly, higher levels of cannabinoid receptors during the light period and decreased expression levels during darkness might be seen in relation to results on a rhythmic expression pattern of the vast majority of hepatic genes driven by food intake [46].

In accordance to this phenomenon, mice showed a bimodal rhythm with a major peak during early evening and a minor peak in the late night [46]. Feeding induced metabolic regulators, which in turn drive rhythmic transcription, in addition to the transcriptional output of the hepatic circadian clock [46]. For example, CB_1_ regulates the metabolic and stress regulators cAMP-response element-binding protein (CREB) and sterol regulatory element binding protein (SREBP).

The activation of CB_1_ in mice increased the hepatic gene expression of the lipogenic transcription factor SREBP-1c and its targets acetyl-CoA carboxylase-1 (ACC) and fatty acid synthase (FAS) [25]. In the same study, treatment with a CB_1_ agonist increased the de novo fatty acid (FA) synthesis in the liver and in isolated hepatocytes. Another study demonstrated a novel mechanism of activated CB_1_, which was able to induce hepatic gluconeogenesis via direct activation of cAMP-responsive element-binding protein H (CREBH) [47]. CREBH functions as a circadian-regulated liver transcriptional regulator that integrates energy metabolism with the circadian rhythm [48]. It was further shown that via CREBH, 2-AG treatment significantly induced hepatic gene expression levels of the key gluconeogenic enzymes PEPCK (phosphoenolpyruvate carboxykinase) and G6pc (glucose-6-phosphatase catalytic subunit) in primary rat hepatocytes, causing a significant increase in glucose production in hepatocytes [49]. It is noteworthy that in mice at the end of the subjective day increased expression of phosphoenolpyruvate carboxykinase 1 (transcript) and Glut-2 (transcript and protein) were observed, and hepatic metabolic genes such as glucokinase and pyruvate kinase showed their peaks of transcripts during the subjective night [50]. Diurnal variations of mRNA levels of lipogenic enzymes were also demonstrated in rat liver [51]. The key lipogenic enzymes fatty acid synthetase (FAS), acetyl-CoA-carboxylase (ACC), and adenosine triphosphate (ATP)-citrate lyase showed diurnal variations by reaching their maximum activity at night and minimum during light periods [52]. Therefore, the diurnal rhythm of cannabinoid receptors is seemingly related to food intake since this is associated with rhythmic expression of the vast majority of hepatic genes of lipid and glucose metabolism [46].

### 2.2. Cannabinoid Receptor CB_1_- and CB_2_-Function and Its Metabolic Consequences in the Liver

Based on the reported increase of both anandamide and CB_1_ receptor levels in the liver it has been suggested, that during an early stage of high-fat-induced obesity, the hepatic ECS can be activated [53]. In fact, there was an increase of CB_1_ receptor levels in the livers of wild-type mice fed with a high-fat diet (HFD) for 3 weeks when compared to wild-type mice fed with a normal diet. CB_1_ receptor-null mice developed significantly less fat mass compared to wild-type mice [26]. An explanation of the underlying mechanism is provided by Liu et al. [54]. The activation of liver CB_1_ receptors by endocannabinoids has been shown to trigger downstream pathways of insulin resistance. One of these target enzymes involved in the progression of insulin resistance is the stearoyl-CoA desaturase-1 (SCD1). It generates monounsaturated fatty acids (MUFAs) which have the ability to inhibit fatty acid amide hydrolase, and by that, promoting increased hepatic levels of anandamide in obese mice (DIO). It is therefore concluded, that the hepatic endocannabinoid/ CB_1_ receptor system is intertwined with SCD1. The upregulation of one, causes a positive feedback loop which ultimately adds to lipogenesis and insulin resistance in obese mice [54]. As a consequence of administering a HFD, increased hepatic fatty acid (FA) synthesis was displayed, as well as the development of a hepatic steatosis phenotype. However, these symptoms were blunted by rimonabant and were absent in CB_1_^−/−^ knockout mice [55]. CB_1_ activation increased FA synthesis in normal mice, an effect that was not observed in CB_1_ knockout mice [56]. The activation of the CB_1_ receptor affects the fat metabolism in the liver by increasing lipogenic gene expression, de novo FA synthesis [25] and lipogenic enzyme activity [57].

Treatment of cultured mouse liver explants with rimonabant increased fat oxidation, whereby, showing that hepatic CB_1_ inverse agonism reduced the expression of genes for prolipogenic enzymes [58].

CB_1_-mediated increase in glycogenolysis and/or gluconeogenesis resulted in increased hepatic glucose production [12]. In line with the previous results, hepatic glucose production was decreased during peripheral infusion of the CB_1_ antagonist rimonabant in diet-induced obese rats [59].

In addition to CB_1_, CB_2_-signaling may also contribute significantly to the mechanisms that regulate lipid accumulation as shown in immortalized human hepatocytes and HepG2 cells [60]. Accordingly, CB_2_ direct agonists may enhance the expression of CB_1_, thereby amplifying the downstream effects on lipid metabolism, resulting in an increased lipid accumulation within the hepatocyte [60]. High-fat diet-induced hepatic steatosis was enhanced in wild-type mice treated with CB_2_ agonist (6a*R*,10a*R*)-3-(1,1-Dimethylbutyl)-6a,7,10,10a-tetrahydro-6,6,9-trimethyl-6*H*-dibenzo[*b*,*d*]pyran (JWH-133) and blunted in CB_2_ receptor knockout mice [39].

### 2.3. Cannabinoid Receptor CB_1_- and CB_2_-Agonism/Antagonism in Metabolic Disorders of the Liver

Steatosis essentially is described as the abnormal retention of lipids in the liver. It is caused by malfunctioning of lipid synthesis, catabolism, and transport from/to the liver. The inflammatory processes involved in steatosis are often also associated with obesity and insulin resistance. Persisting inflammatory processes lead to non-alcoholic steatohepatitis (NASH) and fibrosis, which can progress into more severe stages such as cirrhosis [61]. Chronic overfeeding often results in hepatic lipid accumulation, thereby exceeding the triglyceride storage capacity of hepatocytes. This effect causes the activation of apoptotic and inflammatory pathways, with the development of insulin resistance as a consequence.

Rimonabant, a CB_1_ inverse agonist, has an anti-lipogenic effect in the human hepatocyte cell line (HepG2) by inhibiting liver X receptor-α (LXRα)-dependent SREBP-1c induction. This is mediated by an increase in PKA activity, a protein kinase involved in lipid metabolism, and PKA-mediated liver kinase B1 (LKB1) activation downstream of CB_1_-coupled Gα(i/o) inhibition [62]. Blocking CB_1_ has an impact on carbohydrate and lipid metabolism, which was successfully shown in hepatocytes collected from both lean and obese (*ob*/*ob*) mice. Its effects were the upregulation of genes involved in these metabolisms and interestingly enough, could be reversed when treated with an agonist. It caused a decline in essential hepatic lipogenic proteins [63]. Another study supported these findings and indicated that solely peripherally active CB_1_ inverse agonists/antagonists had therapeutic potential in obesity because of their high efficacy in normalizing the related metabolic abnormalities, including insulin resistance and fatty liver [64].

Obesity-associated hepatic steatosis and dyslipidemia were also significantly reversed in Zucker (*fa*/*fa*) rats after 8 weeks of treatment with CB_1_ specific antagonist, suggesting a direct peripheral effect on carbohydrate and lipid metabolism after blocking CB_1_ in the liver. In addition, chronic treatment with rimonabant reduced obesity-associated hepatic steatosis [65].

Administration of CB_1_ antagonists to obese mice showed favorable effects on the phenotype of liver steatosis, including lipid parameters. The improvement of visceral adipose tissue metabolism via CB_1_ antagonism was reported as a determining factor for the normalization of plasma parameters and the reversion of liver steatosis [66]. In liver explants obtained from both lean and ob/ob mice, blocking the CB_1_ improved both carbohydrate and lipid metabolism indicating a strong differential influence of hepatic CB_1_ receptor [58,66]. This has been supported by Liu et al. [67] who have reported that endocannabinoids contributed to diet-induced insulin resistance in mice via hepatic CB_1_-mediated inhibition of insulin signaling and clearance. Furthermore, CB_1_ activation was involved in the pathogenesis of obesity-related hypertriglyceridemia which underscored the potential efficacy of CB_1_ antagonists in treating metabolic disease [68].

Using a diet-induced obesity (DIO) mouse model, CB_1_ receptor antagonist SR14176 treatment over 5 weeks induced a transient reduction of food intake and a noticeable but sustained reduction of body weight and adiposity. The drug lowered plasma leptin, insulin, and free fatty acid levels. The CB_1_ antagonist SR141716 did not exert the same effects in CB_1_ knockout mice supporting the observation of crucial CB_1_ receptor activity [56]. Indeed, hepatic CB_1_ receptor is involved in development of diet-induced steatosis, dyslipidemia, and insulin and leptin resistance [69]. Notably, CB_1_^−/−^ mice were resistant to HFD [25,56], whereas liver-specific CB_1_ knockout mice developed obesity in similar pattern and intensity as wild-type mice [69]. However, the liver-specific CB_1_ knockout mice showed less steatosis, hyperglycemia, dyslipidemia, and insulin and leptin resistance compared to wild-type mice fed with a HFD. These findings indicated that endocannabinoid mediated activation of hepatic CB_1_ receptors contributes to the diet-induced steatosis and associated hormonal and metabolic changes, but not to the increase in adiposity, observed with high-fat diet [69].

Interestingly, aging seems to have an influence on the cannabinoid receptor-mediated effects [70]. CB_1_-antagonism improved glucose tolerance and enhanced liver insulin sensitivity in aged, but not young, adult mice. A key role for CB_1_ in age-related insulin resistance and metabolic dysfunction has been highlighted, with CB_1_ blockade as a potential strategy for combating metabolic disorders associated with aging [70].

It should be mentioned, that both CB_1_ and CB_2_ played a role in the proper functioning of the lipid metabolism in human-derived immortalized hepatocytes. Genes targeted by both receptors included essential enzymes involved in lipid synthesis and transport [60]. Knocking out the gene for CB_2_ resulted in improved insulin sensitivity when investigating age-related or diet-induced insulin resistance. Whereby, demonstrating an important role of CB_2_ in glucose metabolism by modulating skeletal muscle insulin sensitivity and the inflammatory response in the adipose tissue [71].

Taken together, animal studies as well as human studies have suggested that especially blocking CB_1_ in the liver has a direct peripheral effect on carbohydrate and lipid metabolism (Figure 1). Fatty liver is strongly associated with insulin resistance, increased hepatic glucose production and impaired glucose tolerance in humans [72,73]. Rimonabant was the first, and up until now, the only agent to be approved for the control of obesity because of its body-weight-reducing effects in laboratory animals and man [74]. It was initially considered as a novel therapeutic option for NASH [75], by decreasing fatty liver, increasing hepatic insulin sensitivity, and decreasing hepatic glucose production [76]. However, rimonabant was removed from the market shortly after its introduction in Europe because of severe psychiatric side effects [74]. The development of non-centrally active specific antagonists for CB_1_ isoforms and additionally selective for hepatocytes [40] would be an attractive alternative for the treatment of metabolic liver diseases.

## 3. Other Hepatic Cannabinoid Receptors in Metabolism and Metabolic Disorders of the Liver

### 3.1. G Protein-Coupled Receptor 55 (GPR55)

In addition to CB_1_ and CB_2_ receptors, several deorphanized GPCRs are present in the liver. Recent studies revealed that some cannabinoids and non-cannabinoid ligands bound to the protein of G protein-coupled receptor 55 (GPR55) [77,78,79,80] which might act as a novel “type-3 (CB3)” cannabinoid receptor [81], despite the fact that this receptor does not appear to share a similar fingerprint with any of the classical cannabinoid receptors [82].

GPR55 belongs to the group δ of the rhodopsin-like (class A) family of GPCRs and showed low sequence identity to both CB_1_ and CB_2_ [5,78,83]. GPR55 is suspected to be a possible CBR. It occurs in various tissues regulating energy homeostasis, such as the hypothalamus, gastrointestinal tract, pancreas, liver, white adipose tissue, and the skeletal muscle. Recent studies have demonstrated GPR55 mRNA and protein expression in the liver of rodents and humans [84,85,86]. Accordingly, lysophosphatidylinositol (LPI), an endogenous ligand of GPR55, was abundantly found in the periphery, e.g., high absolute amounts of the major LPI species were quantified in the liver, suggesting an important role in activating GPR55 for the corresponding organ [87,88].

However, the (patho)physiological role of GPR55 in cell dysfunction is still poorly understood, mainly because of the limited identification of downstream signaling targets [86]. Different agonists for GPR55 activated different signaling pathways [86], however, the signaling pathways of this receptor in liver tissue are unknown until now. A current study has shown that GPR55 deficiency in mice was associated with increased obesity, reduced physical activity and energy expenditure. Adipose tissue, liver, and skeletal muscle exhibited a significant reduction in insulin signaling capacity, which may be a consequence of tissue-specific changes in phosphatase and tensin homolog (PTEN) and insulin receptor substrate 1(IRS-1) expression [89]. In accordance, central and peripheral administration of GPR55 agonist O-1602 stimulated food intake in the short-term and increased obesity in the long-term [90]. Another study indicated that the LPI/GPR55 system could be positively associated with obesity in humans [85].

Interestingly, it was further demonstrated that GPR55- and CB_1_-receptor signaling was modulated if receptors are co-expressed, forming a heteromer in vitro. GPR55 signaling was inhibited in the presence of CB_1_ receptors. In contrast, CB_1_ receptor-mediated signaling was enhanced if GPR55 was co-expressed [77]. GPR55 has also been demonstrated to interact with CB_2_ in immune cells [91]. Such crosstalk between GPR55 and CB_2_ signaling might also occur in tissues where both CB_2_ and GPR55 are co-expressed in metabolically active tissues, including the liver, which could have implications for hepatic glucose metabolism [86]. Subsequent high throughput assays have identified GPR55 ligands to not belong to the family of cannabinoids. They also did not bind to either CB_1_ or CB_2_ [92].

### 3.2. G Protein-Coupled Receptor (GPR119)

GPR119 belongs to the group of deorphanized GPCRs and has, although phylogenetically related to CBRs, shown to interact solely with fatty acid amides [5,93,94]. It is present (mRNA and protein) in human and mouse liver tissue as well as in primary mouse hepatocytes [95]. Increased levels of key lipogenic enzymes (FAS, ACC, and SCD1), but not of gluconeogenic enzymes phosphoenolpyruvate carboxykinase and pyruvate carboxylase (PEPCK and PC, respectively—), were observed in hepatocytes isolated from GPR119 knockout mice compared to wild-type mice. In hepatocytes, GPR119 regulated the hepatic lipogenesis by stimulation of 5’-adenosine monophosphate-activated protein kinase (AMPK) phosphorylation [95]. AMPK phosphorylation suppressed SREB-1 expression, which in turn reduced the expression of lipogenic enzymes, including FAS, ACC, and SCD1, thereby inhibiting hepatic lipid accumulation. Conclusively, ligands of GPR119 attenuated hepatic steatosis by inhibiting SREBP-1-mediated lipogenesis [95,96]. Recent results have demonstrated strong in vivo evidence for protection against development of fatty liver by GPR119 agonist APD668 (4-[[1-[2-fluoro-4-(methylsulfonyl)phenyl]-1H-pyrazolo[3,4-d]pyrimidin-4-yl]oxy]-1-piperidinecarboxylic acid,1-methylethyl ester) in a mouse model of NASH. This drug was able to reduce metabolic risk factors such as increased levels of circulating cholesterol, glucose, and triglyceride levels as well as hepatic injury markers. Once fully tested, APD668 might be used in the treatment of non-alcoholic fatty liver disease (NAFLD) or NASH in the future [97].

In mice fed a high trans-fat diet to induce steatohepatitis, monotherapy with either APD668 or linagliptin caused a reduction in the levels of alanine aminotransferase, aspartate aminotransferase, glucose, cholesterol, and epididymal fat mass. These effects were more pronounced upon treatment with a combination of both drugs. On the other hand, combined treatment of APD668 and linagliptin showed a non-significant additive effect in reduction of hepatic triglyceride [98]. Figure 2 gives a summary of GPR55 and GPR119 functions.

## 4. Peroxisome Proliferator-Activated Receptors (PPARs)

PPARs are highly important transcription factors that are activated by numerous synthetic and endogenous ligands. Once activated, they regulate the transcription of downstream target genes [99]. Nuclear receptors, activated under physiological and pathological conditions, are considered as the new target for endocannabinoids [100]. Some cannabinoids activate the different isoforms of PPARs, as shown through the use of reporter gene assays, binding studies, selective antagonists, and knockout studies [101]. PPARs participate in regulating genes involved in fatty acid uptake and oxidation, lipid and carbohydrate metabolism, inflammation, and cell proliferation in general [102,103].

PPARs belong to the family of ligand-activated transcription factors, playing a key role in regulating adipogenesis and inhibiting liver fibrosis [104]. The *PPARα* (NR1C1), *PPARβ*/*δ* (NR1C2), and *PPARγ* (NR1C3) genes share a highly conserved sequence encoding for proteins with high similarity in structure and function [99]. Interestingly, from these known isotypes, each of them has liver cell-type specific patterns of expression (e.g., PPARα, PPARβ/δ in hepatocytes, PPARγ in hepatic stellate cells) [104].

### 4.1. PPARα

PPARα has shown to play a critical role in modulation of energy balance and regulation of hepatic lipid metabolism and acted to reduce hepatic intracellular FA concentrations [105,106]. Indeed, in the liver, PPARα was found mostly in hepatocytes where it prevented triglyceride accumulation [107,108], whereas relative mRNA expression was very low in endothelial cells and Kupffer cells [109]. PPAR*α* was found to stimulate the oxidation of fatty acids on a cellular level in various organelles, such as mitochondria, peroxisomes, and microsomes, as well as the uptake of fatty acids and the synthesis of lipoproteins in hepatocytes. Therefore, it seems to be mainly involved in the fatty acid metabolism [102]. PPARα transcriptionally upregulated numerous genes that are involved in mitochondrial and peroxisomal fatty acid oxidation and in phospholipid remodeling [110,111].

In addition, PPARα affected the duration of an inflammatory response induced by dihydroxy fatty acid leukotriene B_4_ (LTB_4_) [112]. PPARα activation impaired the IL-6 signaling pathway in the liver at the level of membrane receptors and the level of transcription factors [113], and therefore participated in the downregulation of hepatic inflammatory processes.

The main effects of PPARα overexpression or PPAR ligands in the mice liver included a decreased inflammation, increased FA uptake and FA oxidation, decreased very-low-density lipoprotein (VLDL) production, increased high-density lipoprotein (HDL) apolipoproteins, and decreased acute phase reactants [106].

PPARα activated antioxidant enzymes and suppressed hepatic fibrosis in rats [114]. PPARα ligands had an antifibrotic action in the rat thioacetamide (TAA) model of liver cirrhosis, probably due to an antioxidant effect of enhanced catalase expression and activity in the liver. Fatty acid metabolism and ketogenesis are the most conserved PPARα-regulated biological processes in mice and humans, while the glycolysis-gluconeogenesis pathway by PPARα agonists occurred in mice, but not in men [115]. In human hepatocytes, PPARα agonism specifically controlled xenobiotic metabolism and apolipoprotein synthesis pathways. PPARα activation lead to increased β-oxidation rates by fibroblast growth factor 21 (FGF21) activation to provide substrates for ketone body synthesis and gluconeogenesis, thus maintaining energy sources for peripheral tissues [115]. Oleoylethanolamide (OEA) is a natural fatty acid ethanolamide that acted through the activation of PPARα and decreased neutral lipid content in hepatocytes as well as serum cholesterol and triglyceride levels [116]. Further studies have demonstrated that OEA in low and high micromolar concentrations exerted a pharmacological effect by modulating hepatic fibrosis development through the inhibition of hepatic stellate cell (HSC) activation and therefore may be a potential therapeutic agent against liver fibrosis. These results have suggested that endogenous PPARα agonist OEA effectively suppressed activation of HSCs and liver fibrosis through the effects on transforming growth factor (TGF)-β1 [117]. Taken together, studies using mouse models and pharmacological treatments have demonstrated a beneficial effect of PPARα by preventing steatosis, inflammation, and fibrosis. Preclinical as well as clinical studies have shown an impact of PPARα on NAFLD and NASH development [118,119,120,121,122]. PPARα knockout mice fed a HFD seemingly have an increased susceptibility to NASH, as shown by more advanced steatosis and increased markers of oxidative stress and inflammation [118]. Another study implicating a role of PPARα in lipid homeostasis has shown that this nuclear receptor has a sexual dimorphism which causes gender specific alterations in the levels of circulating lipids, in the fat storage and obesity phenotype [119].

Hepatocyte-restricted PPARα deletion in mice showed impaired whole-body fatty acid homeostasis not only during fasting, but also when fed a methionine- and choline-deficient diet or HFD. This PPARα deletion is sufficient in promoting steatosis and hence establishes PPARα as a relevant drug target in NAFLD [110].

### 4.2. PPARβ/δ

Hepatic PPARβ/δ expression was shown in rats [123,124], adult humans [125,126,127], and mice [128]. A high mRNA expression was described for rat hepatocytes, endothelial and Kupffer cells [109], as well as in HSCs [129]. The liver was identified as a major PPARβ/δ-responsive organ, because a PPARδ specific agonist suppressed hepatic glucose output and increased glucose disposal [130].

Due to its seemingly critical role in the liver, this pathway could contribute to the ability of PPARβ/δ agonists to alleviate hyperglycemia and improve insulin sensitivity [130].

PPARβ/δ has been shown to govern hepatic glucose utilization and lipoprotein metabolism and an anti-inflammatory role was previously supported [131]. In detail, the primary metabolic influence of PPAR in the liver was shown to act on the carbohydrate and lipoprotein metabolism, since PPARβ/δ deletion lead to a downregulation of numerous pathways of the carbohydrate metabolism. These include the pentose-phosphate pathway, mannose and fructose metabolism, and in particular, glycolysis. Genes in the latter pathway, that were clearly decreased in PPARβ/δ^−/−^ mice, included pyruvate kinase (*Pklr*) and fructose 1,6 bisphosphatase (*Fbp1*). In addition, PPARβ/δ deletion was associated with decreased expression of a number of genes connected with lipoprotein metabolism (*Apoa4*, *Lipg*), very low density lipoprotein receptor (*Vldlr*), and elevated plasma triglyceride levels in PPARβ/δ^−/−^ mice in the fed state [131].

PPARδ activation was further shown to reduce fasting glucose levels in chow- and high-fat fed mice [132]. This effect was accompanied by hepatic glycogen and lipid deposition as well as up-regulation of glucose utilization and de novo lipogenesis pathways. It was therefore suggested, that PPARβ/δ controls the hepatic energy substrate homeostasis by coordinated regulation of glucose and fatty acid metabolism, which provides a molecular basis for developing PPARδ agonists in order to manage hyperglycemia and insulin resistance [132].

### 4.3. PPARγ

Two isoforms of PPARγ protein were described (PPARγ1 and PPARγ2) and gene products actually generated three mRNAs with the PPARy1 and 3 encoding PPARγ1 protein [61].

PPARγ was found to be strongly expressed in adipose tissue, but human and murine liver contained lower levels [133,134], and equally low transcript levels in rat liver [123]. Recent findings have revealed that in several murine models of obesity and type 2 diabetes mellitus, PPARγ mRNA and receptor proteins are highly up-regulated in the liver. The receptor caused increased transcriptional activity as was demonstrated by the activation of PPARγ-responsive genes in the liver [135,136,137]. This was further supported by up-regulation of the hepatic PPARγ mRNA and protein expression in a diet-induced NAFLD murine model [138].

Treatment with troglitazone, an antidiabetic drug, and thiazolidinone which is targeted by PPAR*γ*, induced the expression of several PPARγ-responsive genes, including aP2, fatty acid translocase (FAT)/CD36, and uncoupling protein-2 (UCP2) in the liver of *ob/ob* mice. Thus, raising the possibility that the effects of PPARγ agonists on lipid metabolism and energy balance may partly be mediated through their effects in the liver [137]. PPARγ2 was proposed to act as an inducer of steatosis in hepatocytes, possibly through an induction of pathways regulating de novo lipid synthesis [139]. Accordingly, hepatospecific PPARγ deletion reduced hepatic fat content in mice fed a HFD that developed hepatic steatosis. This further suggests that PPARγ expression in the liver, especially in hepatocytes, can function as a steatogenic inducer gene [138]. In the human non-alcoholic fatty liver, PPARγ2 and corresponding genes involved in fatty acid partitioning and binding, lipolysis, and monocyte/macrophage recruitment and inflammation were up-regulated [140]. PPARγ was up-regulated in the liver of obese patients with NAFLD, which showed positive associations with SREBP-1c mRNA levels or serum insulin levels, reinforcing a lipogenic mechanism to SREBP-1c induction and up-regulating genes that encode for lipogenic proteins in the development of hepatic steatosis [141].

Elimination of PPARγ in the liver of lipoatrophic A-ZIP/F-1 (AZIP) mice had beneficial effects on the hepatic steatosis. However and not entirely favorable, PPARγ elimination also aggravated the pathological conditions of hyperlipidemia, triglyceride clearance, and muscle insulin resistance [142]. In AZIP-mice lacking a functioning liver PPARγ, the hypoglycemic and hypolipidemic effects of rosiglitazone, an anti-diabetic drug, were abrogated, thereby determining the liver as the major site of action in the absence of adipose tissue. The same drug remained effective in wild-type mice lacking PPARγ in the liver. It could be concluded that adipose tissue is the main target site of thiazolidinediones under normal conditions [142]. Wild-type mice lacking the PPARγ gene in the liver exhibited a fat intolerance, increased adiposity and as a result of triglyceride imbalance, developed hyperlipidemia and insulin resistance. Ultimately, these factors led to hepatic steatosis. Liver PPARγ seems to protect other tissues from triglyceride accumulation and insulin resistance [142]. Liver-specific disruption of PPARγ in diabetic mice dramatically decreased hepatic triglyceride and systemically aggravated insulin resistance [143]. PPARγ regulated the expression of lipogenic genes independently from LXR α (nuclear hormone receptors liver X receptor α) and therefore likely contributes to the generation of fatty liver [144]. Findings indicated that PPARγ2 directly increased hepatic de novo lipogenesis and hepatic triglyceride concentrations [145]. Further investigations have suggested that Cd36-mediated FA uptake and monoacylglycerol (MAG) pathway-mediated FA esterification are major targets of hepatocyte PPARγ. Dysregulation of these FA signaling mechanisms could explain in part the protection against steatosis observed after liver (hepatocyte)-specific PPARγ knockdown (aLivPPARγkd) [146]. ALivPPARγkd dramatically reduced Mogat1 expression, which was reflected by an increase in hepatic MAG levels and indicated a reduced monoacylglycerol O-acyltransferase 1 (MOGAT) activity [146]. Another finding demonstrated that PPARγ orchestrated a gene-expression program downstream of the PI3K/Akt2 pathway, favoring aerobic glycolysis, lipogenesis, and pathophysiological growth. For PPARγ, a cell-autonomous regulation of glycolytic isoenzymes hexokinase 2 and M2 isoform of pyruvate kinase was suggested [147].

Taking together, modulation of receptors related to the endocannabinoid system seems to have a strong impact on liver functions, preferentially by influencing the glycolysis and lipid metabolism (Figure 3).

## 5. Endocannabinoid System and Liver Cirrhosis

As shown above, the endocannabinoid system regulates a wide spectrum of the glucose and lipid metabolism in liver tissue and the impairment of this interaction results in various metabolic disturbances. Accumulation of fat and insulin resistance are the main causes of NAFLD. NAFLD can be classified in two separate entities: non-alcoholic fatty liver or steatosis and non-alcoholic steatohepatitis [148]. Steatosis is characterized by accumulation of fat in more than 5% of the hepatocytes. Inflammation and fibrotic tissue define the presence of NASH. In NASH, hepatocyte injury and damage causes inflammation with direct and indirect activation of immune cells. On the one hand these cells include resident macrophage in the liver, i.e., Kupffer cells and HSCs, and on the other hand infiltrating neutrophils, macrophages, T lymphocytes, and dendritic cells. All cells contribute to liver inflammation and production of proinflammatory cytokines and mediators [148,149]. This inflammation subsequently activates HSCs, which are the main cell type producing fibrotic tissue. Activation of HSCs leads to changes in the phenotype of these cells, developing into myofibroblast-like cells. As a consequence of these processes, fibrotic tissue accumulates in the liver with subsequent progression of NASH, leading to cirrhosis.

### 5.1. Endocannabinoids AEA, 2-AG, and PEA

Recently, it is widely accepted that the normal liver contains the endocannabinoids AEA and 2-AG. There are studies showing relatively low levels of AEA and 2-AG [150], although it is also reported that anandamide and 2-AG are present in the liver at levels similar to those in brain [151,152]. Several types of liver cells including hepatocytes, HSCs and vascular endothelial cells are able to produce the AEA and 2-AG under physiological conditions [32,33,153]. The differences in their levels might be a matter of fasting. Data obtained from different animals models of obesity revealed that DIO mice [25] and Zucker rats exhibited basal and food deprivation/refeeding-induced changes in endocannabinoid levels, e.g., significantly higher 2-AG and/or AEA levels in the liver [154].

There are several studies investigating the expression of endocannabinoids in liver diseases. For example, in acute hepatitis, serum anandamide levels were found to correlate with the extent of tissue damage [155]. Another study indicated that serum levels of endocannabinoids are significantly increased in patients with NAFLD, regardless of their body mass index (BMI), and could be correlated with several metabolic parameters and transaminases [156]. Increased serum concentrations of 2-AG and its precursor and breakdown molecule, arachidonic acid were detected, however, with no significant differences in serum AEA levels in both female and male patients with NAFLD [156].

However, peripheral AEA is increased in patients with end-stage liver disease and the upregulation of the endocannabinoid-related molecules, OEA and palmitoylethanolamine, were even greater than that of AEA [157]. In contrast, the 2-AG levels were not affected by cirrhosis. Interestingly, the major source of elevated endocannabinoids in plasma was extrahepatic presumably a general response to inflammatory processes during cirrhosis.

Both endocannabinoids AEA and 2-AG by themselves seem to have antifibrotic properties, partly independent of cannabinoid receptors. The main responsible liver cell type for producing fibrosis is the HSC, which physiologically constitutes about 5–8% of liver cells. Activation of stellate cells is one of the key mechanisms in the development of fibrosis and cirrhosis. When activated, HSCs change their phenotype to myofibroblast-like cells and undergo morphological and functional changes [158,159]. Remarkably enough, HSCs are also one of the main cells to reverse fibrosis and cirrhosis, which is a consequence of less proliferation and quiescence of the HSCs. In this context, AEA has been considered a potential antifibrogenic tool, since it efficiently induced necrosis in activated HSCs and inhibited the HSC proliferation as shown in surgical specimens of healthy human and rat livers [160]. Higher doses of 2-AG also induced apoptosis in activated HSCs through a receptor independent mechanism, suggesting 2-AG may act as an antifibrogenic mediator in the liver by inducing quiescence in HSCs [161,162]. These findings are further supported by in vitro data. Palmitoylethanolamide (PEA) as a natural occurring cannabinoid shows structurally a close similarity to anandamide. Functionally, it binds neither to CB_1_ nor CB_2_ nor abnormal cannabidiol sensitive receptor [163]. PEA in micromolar concentrations is a major activator of PPARs and suppresses the activation of immune cells such as it prevents the degranulation of mast cells, reduces leukocyte migration to extracellular space and inhibits the proinflammatory cytokine release from macrophages [164,165]. PEA seems to aggravate the effects of other endocannabinoids and in turn its effects are reduced after blocking the CB_2_ receptor [166,167].

N-arachidonoyl dopamine, another endocannabinoid, induced dose-dependent cell death in culture-activated primary murine or human HSCs [168]. However, the endogenous cannabinoids AEA and 2-AG are ubiquitous lipid signaling molecules, exhibiting effects that were mediated mostly by activation of both specific receptors CB_1_ and CB_2_ [36].

### 5.2. CB_1_- and CB_2_-Receptor

Based on the low expression of the receptors the endocannabinoid system is quiescent in normal livers. The progression from an initial liver damage to fibrosis and subsequent cirrhosis is a highly complex process and involves all liver cell types. The role of the ESC in this complex process depends on the progress of the disease.

Endocannabinoids and their hepatic cannabinoid receptors are upregulated in NAFLD and in cirrhosis [162,169,170,171]. Interestingly, the action of CB_1_ and CB_2_ receptors are conversely related with decrease of fibrosis by CB_1_ deficiency and increased collagen deposition by CB_2_ deletion [36,172].

CB_1_ receptors are upregulated in the liver of cirrhotic individuals and overexpressed in liver fibrogenic cells [32]. CB_1_ blockade or inactivation has been associated with decreased amounts of fibrosis, reduced expression of TGF-β and decreased amounts of fibrogenic cells [36]. CB_1_-deficient mice exhibited reduced fibrosis after exposure to chronic carbon tetrachloride (CCl_4_) [173]. Accordingly, chronic CB_1_ receptor blockade with AM-251 in the hepatic microcirculation of common bile-duct-ligated cirrhotic rats was associated with decreased hepatic collagen deposition and an activated phospholipase A_2_ (PLA_2_)/eicosanoid cascade in the cirrhotic liver, mediated by the inhibition of hepatic TGF-β_1_ activity [174]. The antifibrogenic potential of CB_1_ antagonism was also confirmed in C57BL/6J mice given a prolonged HFD to induce NASH and fibrosis [175]. In support of this, rats with CCl_4_–induced cirrhosis had reduced liver fibrosis after a 2-week treatment with CB_1_-antagonist rimonabant [176]. In conclusion, endocannabinoids and the CB_1_ receptor are involved in the development of fibrosis and cirrhosis. Subsequent regression of fibrosis can be achieved by the pharmacological blockade of CB_1_ receptor antagonism even in an advanced stage of the disease when cirrhosis has already developed [177]. This effect was associated with the suppression of a series of pro-fibrogenic and inflammatory mediators [177] and was shown in different animal models of fibrosis and cirrhosis, including NASH.

CB_1_ activation has also been implicated in liver regeneration in mice. This is due to the fact that AEA, by acting via CB_1_, regulates the expression of key cell cycle proteins required for mitotic progression in the regenerating liver. Lack of CB_1_ or CB_2_ blockade by rimonabant was followed by a reduced liver regeneration [178].

CB_2_ receptors were shown to be highly up-regulated in the cirrhotic human liver, predominantly in hepatic fibrogenic cells [30]. In the same study, mice with impaired CB_2_ receptors developed enhanced liver fibrosis after CCl_4_ treatment. Accordingly, CB_2_ knockout mice had an enhanced response to fibrogenic stimuli [30]. The administration of the CB_2_ agonist JWH-133 in the CCl_4_ mouse model reduced the extent of liver injury, indicating that CB_2_ receptor activation alleviates CCl_4_-induced hepatitis and accelerates liver regeneration. Thereby, CB_2_ agonists have been identified as potential beneficial hepatoprotective agents, in addition to their antifibrogenic effects [179]. Accordingly, selective activation of hepatic CB_2_ receptors limited the progression of experimental liver fibrosis as shown in reduced hepatic collagen content in rats with pre-existing cirrhosis [180]. CB_2_ receptor stimulation prevented fibrosis progression in CCl_4_-treated rats and by that, displaying reduced hepatic collagen content, improved mean arterial pressure and portal pressure, ameliorated cell viability, and reduced angiogenesis and cell infiltration compared with untreated fibrotic rats [181]. Others demonstrated that CB_2_ receptor activation decreased liver fibrosis by selectively reducing interleukin 17 (IL-17) production of T-helper (Th)17 lymphocytes via a signal transducer and activator of transcription (STAT)5-dependent pathway, and by blunting the proinflammatory effects of IL-17 on its target cells, while preserving IL-22 production [182].

Taken together, rodent studies have indicated that CB_2_ agonism can reduce the progression of liver fibrosis by decreasing the inflammatory infiltrate (Figure 4).

Whereas, in CB_2_-antagonism or in CB_2_^−/−^ mice an increased collagen deposition, liver fat content, and enhanced inflammatory score was described. These data are further supported by human studies. Recently, a critical role for CB_2_ receptor signaling was confirmed in obese children with NAFLD by showing that the presence of the functional variant of the CB_2_ (Q63R) correlates with severity of the liver inflammation and the presence of NASH [183].

### 5.3. Endocannabinoids and Hemodynamics in Cirrhosis

Cirrhosis is the end stage of different liver diseases and is defined by replacement of normal liver tissue by fibrotic tissue. Activated HSCs undergo myofibroblast-like transformation and mainly produce the fibrotic tissue [184,185]. Other morphologically and functionally different fibroblastic cells, i.e., liver myofibroblasts also contribute to the production of fibrotic tissue in the liver [186]. The damage of the hepatocytes and the increase of fibrotic tissue have different consequences in regard to liver function, liver anatomy, and increase of intrahepatic vascular resistance. The decrease of liver function is resulting in lower degradation and less production of substances. One of the most important substance with lower production in cirrhosis is albumin, which is not only less produced but also dysfunctional. The presence of dysfunctional albumin in cirrhosis explains the strong positive reaction of sodium elimination after exogenous albumin administration.

Another consequence of the increased fibrotic tissue and the destroyed anatomy is the increase of intrahepatic vascular resistance and subsequent development of portal pressure. Portal pressure is the main cause for the development of complications in cirrhotic patients such as the development of ascites and variceal hemorrhage [187]. Furthermore, portal hypertension also causes changes in the splanchnic and systemic vascular system. One of the mechanisms involved in the splanchnic vasodilation is an increase in bacterial translocation in the gut and increase of pathogen-associated molecular pattern release [188]. These activate innate pattern recognition receptors and release proinflammatory molecules, like reactive oxygen species and reactive nitrogen species. The release of proinflammatory molecules is one mechanism behind the decrease of the splanchnic vascular resistance. This splanchnic vasodilatation is one of the drivers of systemic vasodilatation and increased cardiac output. Splanchnic and systemic vasodilation lead to activation of different hormonal systems, mainly the renin-angiotensin-aldosterone system. This causes increased levels of angiotensin and aldosterone and leads to retention of sodium and water, which causes a vicious circle that worsens the development of ascites [188].

Based on several studies it is widely accepted that the concentration of nitric oxide (NO) is increased in the splanchnic and systemic circulation in cirrhosis [189]. A higher concentration of NO is caused by increased levels of endothelial nitric oxide synthethase (eNOS) [190]. The enhanced expression and increased activation of eNOS and augmented endothelial NO release in the vessels is mainly caused by a response to flow and shear stress. This increased eNOS-derived NO production could be explained as normal chronic adaptation of the endothelium in response to chronically elevated splanchnic blood flow in portal hypertension [189]. In cirrhotic rats, blockade of CB_1_ reversed the systemic hypotension, reduced the portal pressure and the mesenteric blood flow [170,191]. The mechanisms behind these processes are not completely understood. Studies indicated that both circulating inflammatory cells and endothelial receptors were involved. In fact, monocytes isolated from the blood of cirrhotic rats and patients, but not from normal controls, caused CB_1_-receptor-mediated hypotension when injected into normal rats. This seemed to be mediated by elevated levels of anandamide from cirrhotic monocytes and elevated levels of CB_1_ receptors from cirrhotic patients in hepatic vascular endothelial cells [170].

A likely source of endocannabinoids in cirrhosis could be activated macrophages, in which lipopolysaccharides induced the synthesis of AEA [192,193]. Indeed, AEA levels were elevated in circulating macrophages of cirrhotic rats and patients [192,193]. Together with the increased expression of CB_1_ receptors in mesenteric arteries, the vasodilatory property of increased levels of AEA was even higher [194,195,196]. These findings suggest AEA as a mediator of splanchnic vasodilation in cirrhosis in a NO independent manner [128].

High cardiac output and low systemic vascular resistance define the hyperdynamic circulation of advanced cirrhosis. Despite this high cardiac output, the cirrhotic cardiomyopathy is defined as a decrease in contractility [197]. There is evidence that the CB_1_ receptor was involved in the suppression of cardiac contractility and blockade of the CB_1_ receptor normalized the contractility [198,199]. Besides the suppression of cardiac contractility by CB_1_ receptor activation, increased myocardial levels of AEA in cirrhosis suggested a further involvement of this pathway in cirrhotic cardiomyopathy [199]. Taking together, endocannabinoids and their classical receptors CB_1_ and CB_2_ seem to be involved in the hemodynamic changes seen in cirrhosis.

## 6. Summary and Conclusions

Hepatic lipid and glucose metabolism was previously indicated as a potential target of the metabolic actions of the endocannabinoid system (ECS).

As further shown by different experimental animal and human studies, the pharmacological and genetic modulation of the ECS is able to induce metabolic changes, extracellular remodeling and the progression of fibrosis. These numerous effects state the importance of ECS in the pathophysiology of metabolic liver diseases such as steatosis and subsequent development of fibrosis.

Thus, the first selective CB_1_ receptor antagonist rimonabant was approved for the treatment of obesity. Unfortunately, it was expected to cause centrally mediated neuropsychiatric side effects, which resulted in the withdrawal of this substance from the market. Currently, further peripherally based agonists or antagonist are under investigation in preclinical settings. They aim at unraveling the pathophysiology of the ECS and to find suitable novel therapeutic tools for the treatment of liver diseases.

For this, however, a better identification of specific signaling pathways of each cannabinoid receptor is necessary, including their interactions within a hepatic cell. In addition, the cross-talk between different hepatic cell types and the understanding of their interactions is crucial to further elucidate the role of ECS in the pathogenesis of liver fibrosis and fatty liver disease.

## Figures and Tables

**Figure 1 ijms-20-02516-f001:**
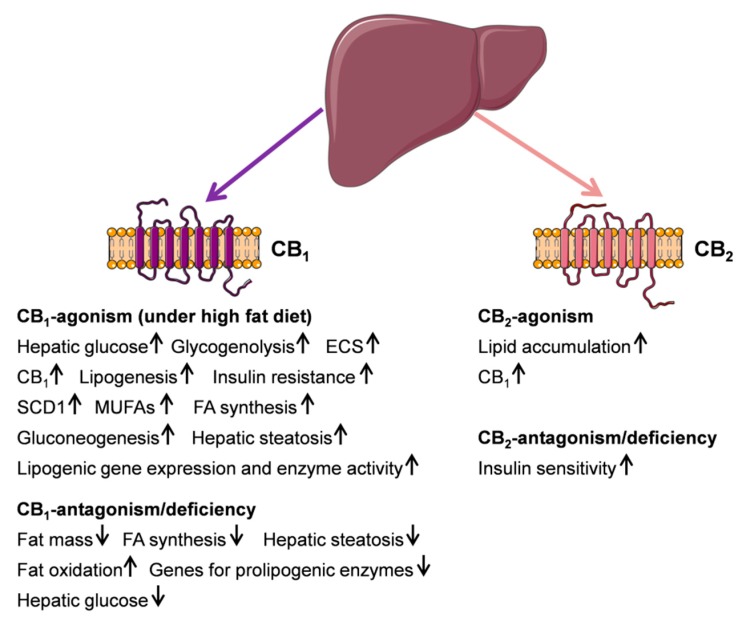
Schematic illustration of cannabinoid receptor (CBR) function in liver metabolism. (↑) and (↓) define up or down regulation. Most findings were generated in rodent models. This figure was prepared using a template on the Servier Medical Art website (Available at: http://smart.servier.com/).

**Figure 2 ijms-20-02516-f002:**
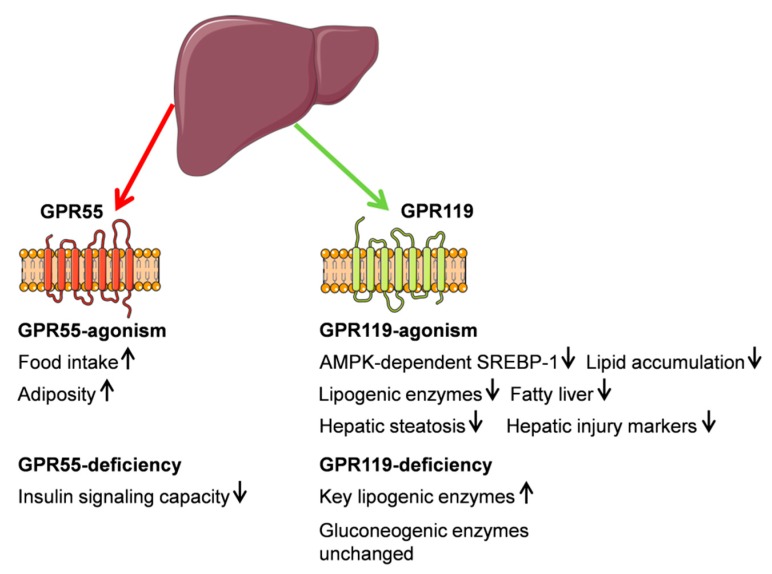
Schematic illustration of G protein-coupled receptor (GPR) function in liver metabolism. (↑) and (↓) define up or down regulation. Most findings were generated in rodent models. This figure was prepared using a template on the Servier Medical Art website (Available at: http://smart.servier.com/).

**Figure 3 ijms-20-02516-f003:**
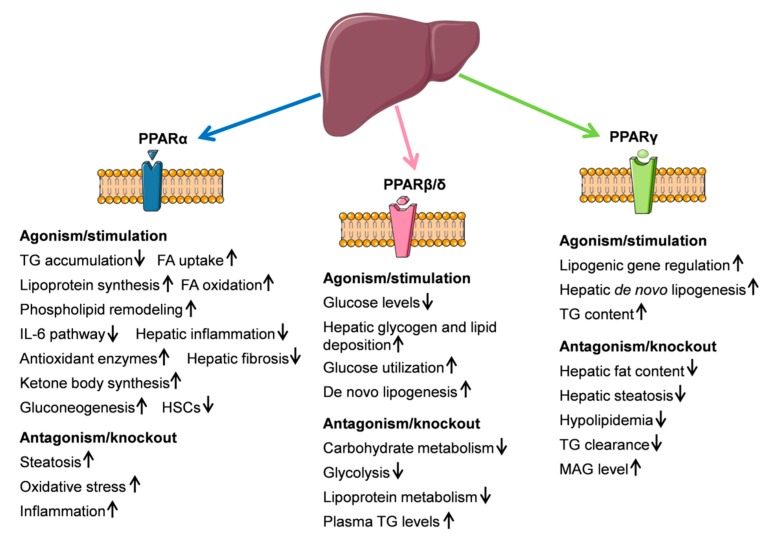
Schematic illustration of peroxisome proliferation-activated receptor (PPAR) function in liver metabolism. (↑) and (↓) define up or down regulation. Most findings were generated in rodent models. This figure was prepared using a template on the Servier Medical Art website (Available at: http://smart.servier.com/).

**Figure 4 ijms-20-02516-f004:**
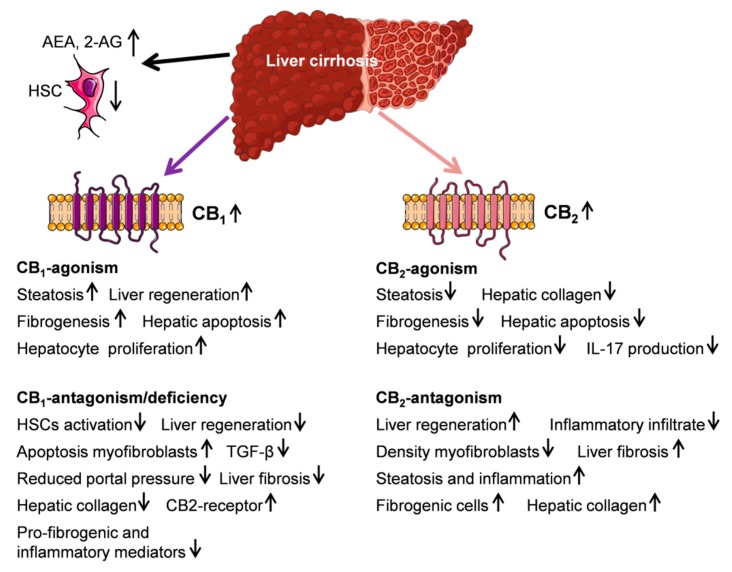
Schematic illustration of cannabinoid receptor (CBR) function in the metabolism of liver cirrhosis. (↑) and (↓) define up or down regulation. Most findings were generated in rodent models. This figure was prepared using a template on the Servier Medical Art website (Available at: http://smart.servier.com/).

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
