# Peer review of "Endocannabinoid System in Hepatic Glucose Metabolism, Fatty Liver Disease, and Cirrhosis"

_ijms, 2019, doi:10.3390/ijms20102516_

Reviewer 1 Report

1. Although this is a potentially interesting review, the manuscript suffers from a number of grave flaws including overtly wrong statements. It is beyond the scope of the review process to indicate all details, since this would involve extensive correction of the manuscript. Just a few hints:

In citing the references the authors fail to consequently distinguish between those effects that were unequivocally attributable to peripheral endocannabinoid actions and those that at least in part might be secondary to central actions.

2. The intracellular signal chain of CB receptors should be described after line 55 on page 2.

3. The authors should state, whether the diurnal expression is maintained in fasted animals

4. Glucokinase is repressed in rat during the light phase and induced in the dark phase during feeding.

5. The impact of CB1 signaling on glucagon action in hepatocytes is not correctly described

6. CB1 action on lipid metabolism is by AMPK not by PKA

7. In their discussion about PPARs the authors lose track of the role of endocannabinoids. They have to more clearly discuss the rank of relevance of endocannabinoids in this system in comparison to other endogenous ligands.

8. In the part on fibrosis/cirrhosis the authors should more clearly describe the role of endocannabinoids on the different cell types.

9. Minor formal mistake:

Apparently part of the introduction was accidentally attached to the abstract (i. e. line 31 onwards). 

Author Response

Reviewer 1:

1. We thank the reviewer for mentioning this important aspect. However and based on the available literature, the consequent differentiation between central and peripheral effects of endocannabinoids on liver function seems extremely difficult when not impossible. Using in vitro systems like hepatic cell lines or primary cultures allow concluding direct endocannabinoid effects on distinct cell types. But it wouldn’t rule out other and presumably opposite effects when central centers become added to the experimental design. There are few experiments performed to separate central and peripheral effects of cannabinoids. For example, Nogueiras et al (2008) administered intracerebroventricularly a CB1 receptor antagonist (Rimonabant) and compared the effects with those injected interaperitoneally. They reported for Rimonabant that low concentrations (3-5µg) are acting more peripherally and high concentrations (10µg) globally i.e. in periphery and central. Furtheremore they reported a reduction in food intake after blocking the central CB1 receptors. Open questions remain, how specific are the observed effects for distinct nuclei when the drug is freely allowed circulating in the cerebrospinal fluid? Especially in hypothalamus there are several nuclei with close vicinity to each other but with different functions. The impact of central sympathetic pathways is poorly investigated as well.

2. Done as requested.

3. In the cited study all animals had access to food ad libitum.

4. We agree in part with the reviewer. In the cited article diurnal (not circadian) levels of liver specific enzymes have been investigated. With light on at 7 O’clock (reported in earlier works of the authors) the mRNA level of Glucokinase reached its maximum at the end of the day. We have now clarified this point in our text.

5. To adequately address this point of criticism, we would kindly ask the reviewer for further information as the impact of CB1 signaling on glucagon action was not handled in the manuscript.

6. As cited in our work, Wu et al. showed the involvement of CB1 signaling on lipid metabolism by influencing the phosphorylation state of PKA. On page 867 of their manuscript it reads “As expected, our data shown here demonstrated the bona fide activation of PKA after rimonabant treatment, which agrees with the reports shown in the experiments using muscle or endothelial cells (Esposito et al., 2008; Huang et al., 2010)”. A very recent publication by Liu et al., Hepatology 2018 referenced to the work by Wu et al. pointing to involvement of AMPK for lipid metabolism.

7. In that part of the discussion the authors focused on cannabinoid receptors and summarized the findings on cannabinoid receptors (CB1 and CB2), other hepatic orphan cannabinoid receptors (GPR55 and GPR119) and Peroxisome proliferator-activated receptors (PPARs). Including endocannabinoids with the knowledge that they are able to simultaneously act via different classical and orphan receptors would distract the discussion from the central theme. We added a general paragraph on PEA to the section 4.1.

8. Based on the low expression of the receptors the ESC is quiescent in normal livers. The progression from an initial liver damage to fibrosis and subsequent cirrhosis is a highly complex process and involves all liver cell types. The role of the ESC in this complex process depends on the progress of the disease. For example, the CB 1 receptor is upregulated in hepatocytes in fatty liver disease, which is one of the initial events in the course of the disease. With progression to fibrosis the CB 1 receptor is upregulated in HSC the main cell type for development of fibrosis and cirrhosis. Furthermore, splanchnic vasodilation occurs in the late phase of the disease and it has been shown that part of this vasodilatation is due to upregulation of the CB 1 receptor in endothelial cells. On the other hand, the CB 2 receptor is expressed in Kuppfer cells (inflammation) and HSC (fibrogenesis). The CB 2 receptor was not detected in hepatocytes so far.

We included this information into the manuscript on page 10, paragraph 4.2.

9. Please apologize our mistake. It has now been corrected.

Reviewer 2 Report

1. This article is well described on endocannabinoid system, CB 1 and CB 2.

2. However, Figures 1-4 are so small in size to see, so they need to be improved.

Author Response

We deeply thank the reviewer for his/her very positive statement. The size of figures 1-4 has accordingly been increased.

Round  2

Reviewer 1 Report

1. This review addresses a very important topic. It contains very much valuable information. The first part describing the cannabinoid system is an excellent introduction into the topic, however, in the attempt to be as comprehensive as possible, the authors lose a bit of focus in the subsequent sections.  There are also some minor inconsistencies. The graphical summaries in the figures are very helpful.

Specific comments:

2. Line 190    Exclude that hepatic effect is secondary to impact on weight gain. See Fig. 1 in citation 66. Or at least discuss this possibility adequately.

3. Line 202  the reasoning is not coherent.  "Indeed, hepatic CB1 receptor is required for development of diet-induced steatosis...... whereas liver-specific CB1 knockout mice developed obesity similar to those of wild-type mice"   please clarify

4. Similarly Line 224 "other receptors of the ECS than CB1 orCB2, do not have these psychotropic effects, which make them attractive targets for the treatment of metabolic liver diseases. "  If  hepatic CB1 receptors are involved (see above) this sentence does not make sense.

5. Lines 228 to 430.   It appears that endocannabinoids affect other GPCRs and PPARs only in high pharmacological concentrations and not in physiological concentrations. From this point of view the discussion of the other GPCRs and PPARs as part of the endocannabinoid system is rather far-fetched.  Since the discussion distracts from the main focus of the review, I would suggest to omit this discussion. At least shorten this part of the manuscript.

Author Response

We are thankful to both reviewers for the time and efforts they had with our manuscript and their constructive suggestion. We addressed the most points of criticisms raised by reviewer 1.

1. We very much appreciate the positive impression of the reviewer.

Specific comments:

2. As requested this point has been discussed in more detail.

3. The reviewer is absolutely right. We have rephrased and clarified this section.

4. We thank the reviewer for pointing out our obvious error. The data has been explained in more detail and inconsistencies omitted.

5. We can only in part agree with the reviewer. The included GPCRs and PPARs are actually discussed in the literature and in the field as novel and putatative cannabinoid receptors. To our today’s knowledge several endocannabinoids such as LPI, OEA or PEA are active agonists of the receptors GPR55 or PPARs with different affinities to its isoforms. It is still a matter of debates which other endocannabinoids are not yet identified activators of these receptors. Additionally, the GPCRs and PPARs play a valuble role in liver physiology and and its pathological events. We considered these aspects as important information for the reader and left this section almost unchanged. As suggested by the reviewer we stressed the micromolar concentrations used in the cited studies.

Reviewer 2 Report

This paper, described on endocannabinoid system, is very well organized.

Thank you for correcting Figures according to my previous peer review comments.

Author Response

We thank the reviewer again for his previous very conscructive criticisms and suggestions helping us to substantially improve our manuscript.